# Unlocking the Power of Co-Occurrence in CLIP: A DualPrompt-Driven Method for Training-Free Zero-Shot Multi-Label Classification

**Ming-Kun Xie**[1], **Zhiqiang Kou**[2]*, **Zhongnian Li**[4], **Gang Niu**[1], **Masashi Sugiyama**[1,3]
[1] RIKEN Center for Advanced Intelligence Project [2] Hong Kong Polytechnic University
[3] The University of Tokyo [4] China University of Mining and Technology
`ming-kun.xie@riken.jp` `{zqiang.kou, gang.niu.ml}@gmail.com`
`zhongnianli@cumt.edu.cn` `sugi@k.u-tokyo.ac.jp`

## Abstract

Contrastive Language-Image Pretraining (CLIP) has exhibited powerful zero-shot capacity in various single-label image classification tasks. However, when applied to the multi-label scenarios, CLIP suffers from significant performance declines due to the lack of explicit exploitation of co-occurrence information. In pretraining, due to the contrastive property of its objective, the model focuses on the prominent object in an image, while overlooking other objects and their co-occurrence relationships; during inference, it uses a discriminative prompt containing only a target label name to make predictions, which does not introduce any co-occurrence information. Then, an important question is as follows: *Do we need label co-occurrence in CLIP for achieving effective zero-shot multi-label learning?* In this paper, we propose to rewrite the original prompt into a correlative form consisting of both the target label and its co-occurring labels. An interesting finding is that such a simple modification can effectively introduce co-occurrence information into CLIP and it exhibits both good and bad effects. On the one hand, it can enhance the recognition capacity of CLIP by exploiting the correlative pattern activated by the correlative prompt; on the other hand, it leads to object hallucination in CLIP, where the model predicts objects that do not actually exist in the image, due to overfitting to co-occurrence. To address this problem, we propose to calibrate CLIP predictions by keeping the positive effect while removing the negative effect caused by suspicious co-occurrence. This can be achieved by using dual prompts consisting of the discriminative and correlative prompts, which introduce label co-occurrence while emphasizing the discriminative pattern of the target object. Experimental results verify that our method can achieve better performance than the state-of-the-art methods.

## 1 Introduction

Vision-Language Models (VLMs) trained on massive cheaply collected data have demonstrated immense potential in many realistic tasks, *e.g.*, object detection (Gu et al., 2022), semantic segmentation (Lin et al., 2023), and anomaly detection (Zhou et al., 2024). As a representative model, CLIP performs pretraining by aligning large-scale image-text using a contrastive objective, showing strong zero-shot generalization ability in various downstream tasks. To perform zero-shot image classification, a vanilla strategy is to expand each label name with a set of prompt templates, *e.g.*, "A photo of a {label}", showing impressive zero-shot classification performance in many benchmark datasets Radford et al. (2021). Although CLIP has achieved successes in the single-label zero-shot classification tasks, it often suffers from unfavorable performance when applied to more realistic multi-label scenarios, where an image is often associated with multiple labels.

---

*Corresponding author

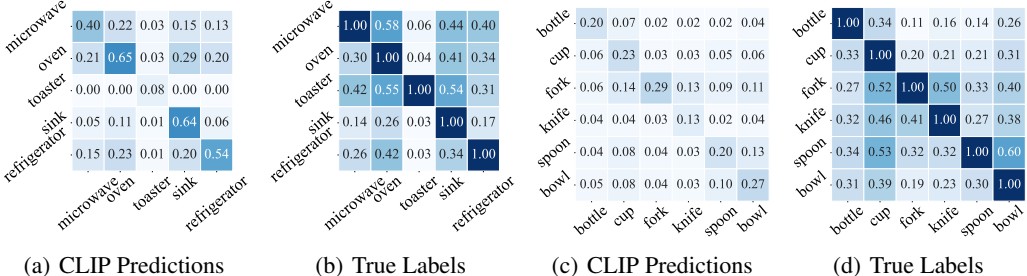

(a) CLIP Predictions      (b) True Labels      (c) CLIP Predictions      (d) True Labels

Figure 1: The co-occurrence probabilities of CLIP predictions and true labels for groups of classes. Without explicitly exploiting label correlations, CLIP is hard to precisely model the co-occurrence among labels.

There are two major challenges in achieving effective zero-shot multi-label classification. As pointed by Lin et al. (2024), the first challenge is that CLIP tends to capture the global features of an entire image dominated by the most prominent object, while neglecting the local features of others. This limits its ability to recognize multiple objects simultaneously, often leading to the precise recognition of only the most prominent object while overlooking others. To address this, TagCLIP (Lin et al., 2024) adopted the Vision Transformer (ViT) (Dosovitskiy et al., 2021) as the backbone of CLIP to explore the local features of patches, which are usually useful for identifying inconspicuous objects. However, one limitation of this method lies in its heavy reliance on ViT to obtain local features of patches for performing classification, which reduces its universality and prevents it from being applied to other backbones, *e.g.*, ResNet (He et al., 2016).

The other challenge is that CLIP does not explicitly leverage label correlations, *i.e.*, co-occurrence relationships, during both pretraining and inference, which has been proven to be indispensable for multi-label classification Chen et al. (2019); Lanchantin et al. (2021). In pretraining, the contrastive objective used in CLIP leads its image encoder to focus on the prominent object in an image (Lin et al., 2024), while neglecting others and their co-occurrence patterns; in inference, using a discriminative prompt such as "A photo of a {label}" does not introduce any co-occurrence information. To verify this, we perform zero-shot multi-label classification on benchmark dataset, MS-COCO (Lin et al., 2014), and show the co-occurrence probabilities of vanilla CLIP predictions and true labels in Figure 1. Considering that it is too large to display the entire matrix in the main text, we select two subsets of frequently co-occurring objects and show their co-occurrence probabilities. From the figure, there is a significant gap between the estimated co-occurrence probabilities and the true ones, which means that vanilla CLIP cannot precisely model co-occurrence relationships. This indicates that it cannot effectively reduce the complexity of the output space by capturing prior information of label relationships, which often leads to missing labels and consequently degrades model performance. An important question has emerged: *Do we need label co-occurrence in CLIP to achieve effective zero-shot multi-label classification?*

In this paper, unlike the previous work that focuses on enhancing image feature representation, we propose to rewrite the original discriminative prompt and obtain a correlative prompt for each target label by including its co-occurring labels. An interesting finding is that such a simple modification effectively introduces co-occurrence information, prompting the model to capture co-occurrence patterns, which enhances its ability to recognize multiple objects. Unfortunately, we find that CLIP tends to overfit to the co-occurrence relationships introduced by the correlative prompts. This leads the model to suffer from an object hallucination issue, where it makes predictions even when the target object does not exist in the image while its co-occurring objects do. To address this problem, inspired by the recent study (Xie et al., 2024), we build a causal inference framework by modeling the co-occurrence as a mediator. This enables us to calibrate the CLIP predictions by removing the biased prediction caused by mediated effect from the prompt of co-occurring labels to target prediction. In practical implementation, our derivation shows that this can be achieved simply by combining the outputs of the discriminative prompt and the correlative prompt. Comprehensive experimental results verify that our method can achieve better performance than state-of-the-art methods.

## 2 RELATED WORKS

Multi-label image classification aims to train a multi-label classifier that can predict all relevant labels for unseen images. The pioneering work (Wang et al., 2016) combined Convolutional Neural Networks (CNNs) and Recurrent Neural Networks (RNNs) and developed a CNN-RNN framework to characterize the label correlations as well as the image-label relevance. To address the issue that dataset-level statistical correlations may not hold for every individual image, Structured Semantic Transfer (SST) (Chen et al., 2022) introduced the Intra-image Semantic Transfer (IST) module, which learns an image-specific co-occurrence matrix and leverages this information to recover unknown labels. To overcome the challenge that incomplete annotations make it difficult to estimate label co-occurrence, SCPNet (Ding et al., 2023) computed text features by feeding category names into the CLIP text encoder, and used their pairwise similarities as a surrogate for the label co-occurrence. Zhang et al. (2024) developed a large model to achieve the goal of recognizing any object. Yue et al. (2024) treated the task of object recognition as the task of next token prediction. Some studies train multi-label classifiers using only a small number of labeled examples (Xie et al., 2023; Xiao et al., 2024; Kou et al., 2025). Other studies transform multi-label information into label distributions to effectively learn classification models (Kou et al., 2024).

Prompt tuning (Zhou et al., 2022) learns continuous vectors as task-specific prompts based on a small number of training examples. The idea has been applied to multi-label image classification and achieved impressive performance (Sun et al., 2022; Hu et al., 2023). Given that images may be costly to collect while texts are cheap to generate, Guo et al. (2023) proposed Texts as Images in prompt tuning (TaI) to adapt CLIP to multi-label image classification based on only the textual modality.

## 3 PRELIMINARIES

In our setting, the task is to perform multi-label image classification, also known as object recognition Yue et al. (2024) or image tagging Zhang et al. (2024), using CLIP without training. Specifically, we are given a dataset $\{x_i\}_{i=1}^n$ consisting of $n$ test instances. Each instance is associated with an unknown label vector $y_i \subseteq \mathcal{Y}$, where $\mathcal{Y} = \{0,1\}^q$ is the label space with $q$ possible class labels. Here, $y_k = 1$ indicates that the $k$-th label is relevant while $y_k = 0$ indicates that it is not. Our task is to develop an inference strategy that enables CLIP to accurately predict all relevant labels for each test instance. We begin with a brief introduction of CLIP. We use $[q]$ to denote the integer set $\{1, \ldots, q\}$.

CLIP consists of an image encoder $\text{Enc}_\text{I}(\cdot)$ and a text encoder $\text{Enc}_\text{T}(\cdot)$. To perform zero-shot classification, a conventional method is to construct a prompt $P_k$ using a template like "`A photo of a {label`$_k$`}`", where `label`$_k$ is the name of the $k$-th label. By using these two encoders, the image and text features can be obtained as

$$\boldsymbol{f}_i = \text{Enc}_\text{I}(\boldsymbol{x}_i), \quad \boldsymbol{t}_k = \text{Enc}_\text{T}(P_k), \forall k \in [q].$$

Then, by computing the cosine similarity between the image features $\boldsymbol{f}_i$ and the text features $\boldsymbol{t}_k$ and applying the *softmax* activation function, we obtain the predicted probabilities as

$$p(y_k = 1 | \boldsymbol{x}_i, P_k) = \frac{\exp(\langle \boldsymbol{f}_i, \boldsymbol{t}_k \rangle)}{\sum_{k=1}^q \exp(\langle \boldsymbol{f}_i, \boldsymbol{t}_k \rangle)}, \forall k \in [q],$$

where $\langle \cdot, \cdot \rangle$ denotes the cosine similarity.

The original paper of CLIP has shown that prompt engineering and ensembling can enhance the performance of its zero-shot classification (Radford et al., 2021). For example, the original single prompt can be expanded into a set of prompts by including prompts such as "`A satellite photo of a {label}`" and "`A photo of a big {label}`".

## 4 DO WE NEED CO-OCCURRENCE IN CLIP?

As discussed before, a potential limitation of CLIP is that it does not explicitly exploit co-occurrence information during either the pretraining phase or the inference phase. Co-occurrence information

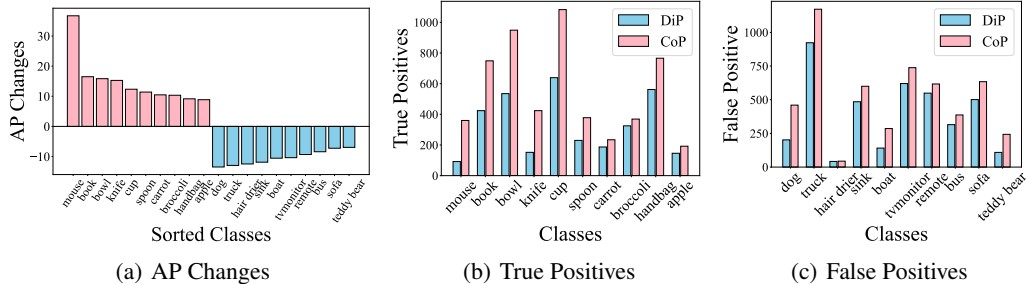

(a) AP Changes       (b) True Positives       (c) False Positives

Figure 2: Empirical Validations on the advantages and disadvantages of CoP. (a) shows top 10 performance improvements and top 10 performance declines after using CoP. (b) shows the co-occurrence probabilities of CLIP predictions using CoP. (c) reports the number of false positives that contain co-occurring objects predicted by DiP and CoP.

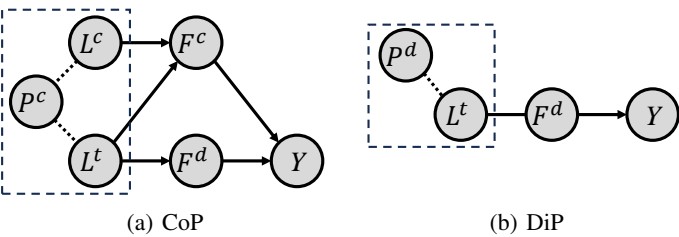

(a) CoP                           (b) DiP

Figure 3: The causal graphs of DiP and CoP. $P^d$ and $P^c$ are the discriminative and correlative prompts. $\mathbb{F}^d$ and $\mathbb{F}^c$ are the discriminative and correlative features. $Y$ is the prediction of the target label.

has long been proven to significantly enhance multi-label classification performance (Chen et al., 2019). Consequently, an important question arises: *Do we need co-occurrence in CLIP for zero-shot multi-label classification?* To answer this question, we perform experiments on MS-COCO by incorporating co-occurrence information during the inference phase.

Specifically, for each label $y_k$ with the name $\texttt{label}_k$, we assume that there is a set of co-occurring labels $\mathcal{C}_k = \{\texttt{label}_j\}_{j=1}^{m_k}$, where $m_k$ is the number of co-occurring labels for the label $y_k$. We will discuss how to obtain the co-occurring label set in the latter section. Then we rewrite the prompt as "A photo of a $\{\texttt{label}_k\}$ often includes a $\{\texttt{label}_1\}$, ..., a $\{\texttt{label}_{m_k}\}$". This allows us to introduce co-occurrence information into CLIP inference through prompts. We refer to this prompt as the Correlative Prompt (CoP), while referring to the original prompt as the Discriminative Prompt (DiP). Figure 2(a) illustrates the performance improvements/declines of CoP over DiP, sorted by the performance changes in terms of commonly used Average Precision (AP). As shown in the figure, some labels experienced significant performance improvements, while others saw substantial declines. Due to the page limit, we only show the top-10 labels with the highest performance improvements and the top-10 with the greatest performance declines. The complete results can be found in Appendix B. To disclose the reason why the improvements are achieved on these labels, we report the number of true positive instances that contain the target label and its co-occurring labels, measuring how large co-occurrence helps the model to identify the target object. The metric measures the positive impact of co-occurrence on the CLIP predictions. Obviously, CoP makes far more correct predictions than DiP on most of these labels. This indicates that co-occurrence is useful for enhancing CLIP's ability to recognize multiple objects.

In Figure 2(a), we also present top-10 labels with the greatest performance declines achieved by CoP. To provide an explanation for this phenomenon, for each target label, Figure 2(c) illustrates the number of false positive instances that contain its co-occurring labels, measuring how many incorrect predictions are caused by the model's overfitting to co-occurrence. It is obvious to see that CoP makes far more incorrect predictions caused by the overfitting issue than DiP in almost all

labels. While CoP benefits from co-occurrence information, it still carries the risk of overfitting to it.

**A Causal Perspective** To provide a theoretical explanation of these phenomena, as shown in Figure 3(a), we construct a causal inference framework to show how CoP leverages co-occurrence information. The framework includes three kinds of variables: the correlative prompt $P^c$ consisting of the target label $L^t$ and its co-occurring labels $L^c$, the discriminative activated features $F^d$ and the correlative activated features $F^c$, and the model prediction $Y$ for the target label. We use $\rightarrow$ to represent the causal link between any two variables, which indicates their causal effects. For the sake of comparison, Figure 3(b) shows the causal graph of DiP, where $P^d$ represents the discriminative prompt only containing the target label $L^t$. On the one hand, compared to DiP, which can only make predictions by activating the discriminative features through the single path $L^t \rightarrow F^d \rightarrow Y$, CoP can still make predictions by activating the correlative features through an additional path $(L^t, L^c) \rightarrow F^c \rightarrow Y$, especially when the former does not work. This improves CLIP's ability to recognize inconspicuous objects by leveraging co-occurrence information, which is verified by the results in Figure 2(b). On the other hand, we consider a case where the target object does not exist in an image but its co-occurring objects do. The discriminative features would not be activated by the target label $L_t$ in the correlative prompt, *i.e.*, the path $L^t \rightarrow F^d \rightarrow Y$ is disconnected, due to the fact that there is no target object; while the correlative features would be activated by the co-occurring labels $L^c$ in the correlative prompt, *i.e.*, the path $L^c \rightarrow F^c \rightarrow Y$ is connected, due to the fact that there exist co-occurring objects. This enables the model to give a high probability of the target object that does not even exist, which can be supported by the results in Figure 2(c). The above discussions tell us that co-occurrence has not only a positive side: it enhances the CLIP's ability to identify the target objects, but also a negative side: it makes CLIP suffer from the overfitting issue and causes object hallucination (Biten et al., 2022). The important question now becomes: *Can we keep the positive side while mitigating the negative side of co-occurrence in CLIP?*

## 5   UNLOCKING THE POWER OF CO-OCCURRENCE

For CoP, the predicted probability can be written as $p(y_k = 1|\boldsymbol{x}_i, P_k^c)$ or $p(y_k = 1|\boldsymbol{x}_i, L_k^t, L_k^c)$, where $P_k^c$ is the correlative prompt consisting of the target label $L_k^t$ and its co-occurring labels $L_k^c$ for the $k$-th class. However, when there are only co-occurring objects in an image and no target object, CLIP still activates the correlative features by the co-occurring labels in the prompt through the path $L^c \rightarrow F^c \rightarrow Y$ and gives a high probability of the target object. This indicates that CLIP suffers from overfitting to the co-occurrence, leading to object hallucination. To address this problem, we propose to calibrate the CoP prediction via the total direct effect (TDE) (Pearl, 2001), with the goal of removing the biased part of the prediction. For a given image $\boldsymbol{x}$, we define the TDE prediction with respect to the $k$-th class as

$$\mathcal{T}_k(\boldsymbol{x}) = p(y_k = 1|\boldsymbol{x}, L_k^t, L_k^c) - p(y_k = 1|\boldsymbol{x}, L_k^c). \tag{1}$$

The first term represents the positive causal effect, where CLIP makes a prediction based on the target label prompt $L^t$ and its co-occurring label prompt $L^c$; the second term represents the negative effect, where CLIP makes a prediction based on only the co-occurring label prompt. The minus sign indicates that we keep the positive effect while discarding the negative effect.

However, in our experiments, we find Eq. (1) hardly works. This may be because CLIP often overestimates the probability $p(y_k = 1|\boldsymbol{x}, L_k^c)$, especially when some labels preferred by CLIP appear in $L^c$. This leads the final predictions to underestimation, resulting in performance declines. To address this problem, we derive an equivalent form of Eq. (1), which transforms it from a subtraction form into an addition form as [1]

$$\mathcal{T}_k(\boldsymbol{x}) = p(y_k = 1|\boldsymbol{x}, P_k^c) + p(y_k = 1|\boldsymbol{x}, P_k^d). \tag{2}$$

The second term represents the direct causal effect, where CLIP makes a prediction based on the discriminative prompt $P^d$ containing only the target label $L^t$. The plus sign indicates that we enhance the direct effect. An intuitive explanation of this transformation is that mitigating the indirect effect is equivalent to strengthening the direct effect. We refer to this method as DualPrompt.

---

[1]The detailed derivation can be found in Appendix A.

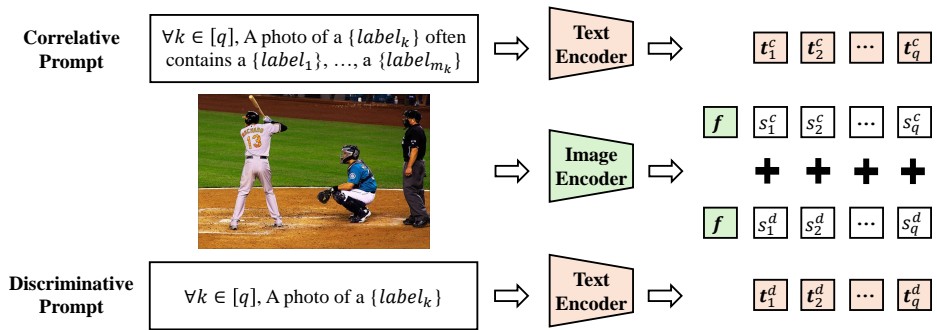

Figure 4: The framework of DualPrompt. It consists of two types of prompts. The correlative prompt is used to maintain the good causal effect of co-occurrence by exploring correlative patterns, while the discriminative prompt is used to remove the bad causal effect by enhancing the discriminative responses.

Figure 4 provides an illustration of our proposed DualPrompt method. Unlike the vanilla method, the bottom branch, which uses the discriminative prompt, our DualPrompt method uses dual prompts, a discriminative prompt and a correlative prompt, to achieve effective zero-shot multi-label classification.

**How to Obtain Co-Occurrence?** Considering that co-occurrence has both universality, meaning that it is a natural law used to describe the physical world, and specificity, meaning that it may vary across different datasets, we employ two methods to obtain co-occurrence information in experiments. Firstly, we ask ChatGPT-4o to generate up to $l$ labels that frequently appear with target objects in the same images. We find that the co-occurring labels generated by ChatGPT are not completely correct. A large $l$ often means that more labels without co-occurrence relationships are introduced, which can mislead the model and result in performance declines. To avoid this problem, we set $l$ to be as small as 2 in our experiments. The other method is to annotate a tiny amount of training data and only use it for estimating the co-occurrence probabilities. The most co-occurring labels for each target label can be found according to the co-occurrence probabilities. Our empirical studies in Section 6 show that a very small number of examples, *e.g.*, $1\%$ training data in the MS-COCO dataset, is needed to obtain an acceptable co-occurrence probability, which are often insufficient to fine-tune CLIP by existing methods (Sun et al., 2022) to achieve favorable performance.

## 6 EXPERIMENTS

We first compare our proposed method with the state-of-the-art methods; then, we perform ablation studies to examine the contribution of each component.

### 6.1 EXPERIMENTAL SETTINGS

**Dataset** To evaluate the performance of the proposed method, we perform experiments on two benchmark datasets, including MS-COCO 2014 [2] (MS-COCO for short) (Lin et al., 2014) and Visual Genome [3] (Krishna et al., 2017). MS-COCO contains 82,081 training images and 40,137 validation images for 80 classes, with an average of 2.9 labels per image. Visual Genome is a dataset that contains 108,249 images from 80,138 categories. By performing preprocessing as done in the previous work (Xie et al., 2024), we obtain a dataset named VG-256 that contains 106,702 images and 256 classes. We randomly split the entire dataset into a 70% training set with 74,691 images and a 30% validation set with 32,011 images. We directly evaluate our method and other methods on the validation set. The commonly used mean Average Precision (mAP) and per-class F1 (F1) are used as the evaluation metrics. To ensure a fair comparison, we set the random seed to 1 for all experiments.

---

[2] https://cocodataset.org
[3] https://homes.cs.washington.edu/~ranjay/visualgenome/index.html

Table 1: The comparative results between the proposed method and the state-of-the-art methods on MS-COCO dataset. The best performance is highlighted in bold. Coo. Est. represents co-occurrence probabilities estimation.

| | Backbone | Resolution | Training Data Usage | mAP | F1 |
|---|---|---|---|---|---|
| DualCoOp | ResNet-101 | $224 \times 224$ | 1% Data for Training | 56.3 | 55.1 |
| TaICLIP | ResNet-101 | $224 \times 224$ | 1% Data for Training | 56.6 | 55.7 |
| TaICLIP | ResNet-101 | $224 \times 224$ | COCO Captions | 65.1 | – |
| CLIP | ResNet-101 | $224 \times 224$ | None | 62.9 | 59.8 |
| DualPrompt | ResNet-101 | $224 \times 224$ | None | 65.5 | 61.7 |
| DualPrompt | ResNet-101 | $224 \times 224$ | 1% Data for Coo. Est. | 67.1 | 63.0 |
| DualCoOp | ViT-B/16 | $224 \times 224$ | 1% Data for Training | 55.1 | 54.4 |
| TaICLIP | ViT-B/16 | $224 \times 224$ | 1% Data for Training | 63.6 | 55.9 |
| CLIP | ViT-B/16 | $224 \times 224$ | None | 64.9 | 61.5 |
| DualPrompt | ViT-B/16 | $224 \times 224$ | None | 67.7 | 63.6 |
| DualPrompt | ViT-B/16 | $224 \times 224$ | 1% Data for Coo. Est. | 69.4 | 65.0 |
| TagCLIP | ViT-B/16 | Original | None | 68.7 | 65.2 |
| DualPrompt + TagCLIP | ViT-B/16 | Original | None | 69.2 | 65.4 |
| DualPrompt + TagCLIP | ViT-B/16 | Original | 1% Data for Coo. Est. | **70.0** | **66.1** |

Table 2: The comparative results between the proposed method and the state-of-the-art methods on VG-256 dataset. The best performance is highlighted in bold. Coo. Est. represents co-occurrence probabilities estimation.

| | Backbone | Resolution | Training Data Usage | mAP | F1 |
|---|---|---|---|---|---|
| DualCoOp | ResNet-101 | $224 \times 224$ | 2% Data for Training | 30.4 | 33.5 |
| TaICLIP | ResNet-101 | $224 \times 224$ | 2% Data for Training | 32.0 | 34.9 |
| CLIP | ResNet-101 | $224 \times 224$ | None | 29.2 | 32.2 |
| DualPrompt | ResNet-101 | $224 \times 224$ | None | 33.5 | 36.1 |
| DualPrompt | ResNet-101 | $224 \times 224$ | 2% Data for Coo. Est. | 35.2 | 37.6 |
| TagCLIP | ViT-B/16 | Original | None | 39.6 | 41.9 |
| DualPrompt + TagCLIP | ViT-B/16 | Original | None | 40.2 | 42.4 |
| DualPrompt + TagCLIP | ViT-B/16 | Original | 2% Data for Coo. Est. | **40.7** | **42.7** |

**Implementation** For CLIP, we use ResNet-101 (He et al., 2016) and ViT-B/16 (Dosovitskiy et al., 2021) as the backbones. For TagCLIP, we use the original resolution of images as done in its original paper; for other methods, we use a resolution of $224 \times 224$. Following the previous work (Lin et al., 2024), we use the 80 prompts used in CLIP (Radford et al., 2021). For prompt fine-tuning methods, when only 1% of the training data is available, the batch size is set to 16; when 2%-5% of the training data is available, the batch size is set to 32. The other parameters use the reference values provided in the original papers. All experiments were run on a single NVIDIA A100 GPU.

## 6.2 COMPARISON WITH STATE-OF-THE-ART METHODS

To validate the effectiveness of the proposed method, we compare it with two types of CLIP-based methods: i) two training-based methods, DualCoOp (Sun et al., 2022), which performs prompt tuning based on a subset of downstream training data, and TaI (Guo et al., 2023), which trains on a subset of data or curated caption data in the downstream task; ii) two training-free methods, CLIP (Radford et al., 2021) and TagCLIP (Lin et al., 2024).

Table 1 and Table 2 report the results of the proposed method and other methods on MS-COCO and VG-256, respectively. From the table, we can see that: i) DualPrompt outperforms both training-based and training-free methods by a significant margin. ii) By combining DualPrompt and Tag-

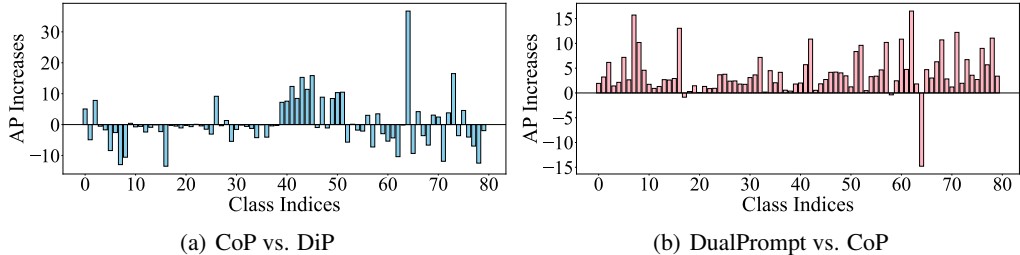

(a) CoP vs. DiP  (b) DualPrompt vs. CoP

Figure 5: Comparisons of per-class average precision among DiP, CoP, and DualPrompt. Compared with DiP, CoP shows improvements on some categories but decreases on others, whereas compared with CoP, DualPrompt achieves improvements on the vast majority of categories.

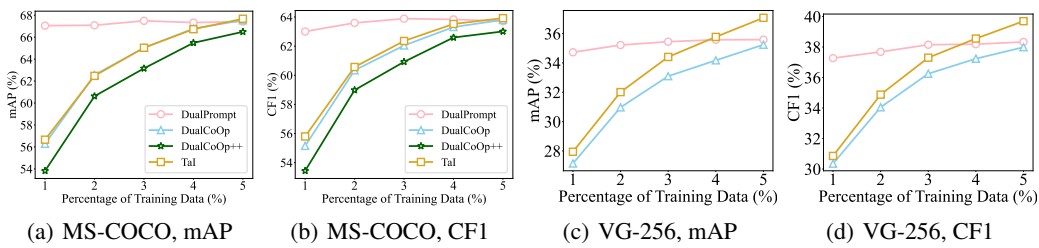

(a) MS-COCO, mAP  (b) MS-COCO, CF1  (c) VG-256, mAP  (d) VG-256, CF1

Figure 6: The performance curve with the increase of training data usage. For DualPrompt, training data is only used for estimating co-occurrence probabilities; while for DualCoOp, training data is used for model training.

CLIP, we achieve the state-of-the-art performance on both datasets. iii) For the two methods of obtaining co-occurrence labels, it is evident that co-occurrence estimation with a tiny amount of data is better than ChatGPT. The co-occurrence provided by ChatGPT is universally existing regularity, while co-occurrence estimation is based on the prior of the downstream dataset. These results convincingly verify that the proposed method can achieve the state-of-the-art performance.

## 6.3  ABLATION STUDIES

We provide empirical validations on how DualPrompt keeps the positive impact while removing the negative impact of co-occurrence. Figure 5 illustrates a stepwise evaluation on MS-COCO in terms of AP. From the figure, we can see that CoP achieves better performance than DiP on some classes, while performing worse on others. Overall, the performance of CoP is inferior to that of DiP. This is because CLIP overfits to the co-occurring labels given in the prompt, making many false positive predictions, which leads to a decline in model performance. To address this problem, our proposed method calibrates the predicted probabilities via TDE measurement by removing the biased parts. DualPrompt achieves performance improvements in almost all classes, providing convincing evidence that it can maintain the positive effect while removing the negative effect.

## 6.4  STUDY ON TRAINING DATA USAGE

In our method, we need to obtain the co-occurring labels for each target label. A solution is to estimate the co-occurrence probabilities based on a very small subset of training data and obtain co-occurrence according to probabilities. In this subsection, we show that such a small number of training data is insufficient for model training. Figure 6 illustrates the performance curves of our method and prompt tuning methods, DualCoOp and TaI. It is noteworthy that our method only uses the training data for estimating co-occurrence probabilities, while DualCoOp and TaI use them for fine-tuning the model. From the figure, it can be observed that with a very small number of training (1%), there is a significant performance gap between our method and prompt tuning methods. With the increase of training data, the gap gradually narrows, and eventually, TaI outperforms our

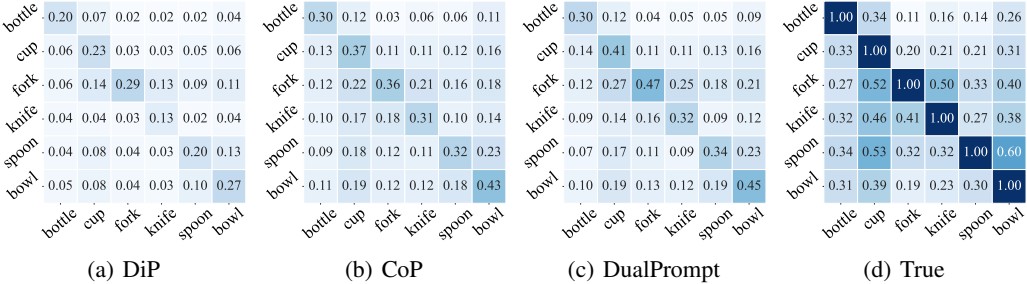

|  | (a) DiP | (b) CoP | (c) DualPrompt | (d) True |

Figure 7: The co-occurrence probabilities of different predictions and true labels.

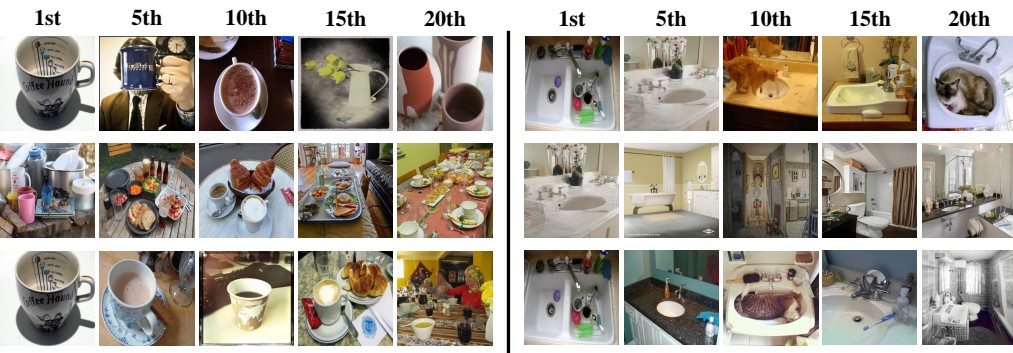

Figure 8: The text-image retrieval results of DiP (the first row), CoP (the second row) and Dual-Prompt (the last row) with respect to *cup* (on the left side) and *sink* (on the right side). To show the representativeness of different methods, we present the images ranked 1st, 5th, 10th, 15th, and 20th based on text-image similarities.

method. This indicates that our method requires such a small number of training data to estimate co-occurrence probabilities that they are not even sufficient for fine-tuning the model.

## 6.5 STUDY ON CO-OCCURRENCE ESTIMATION

In this subsection, we demonstrate the ability of different methods to capture co-occurrence relationships. Figure 7 illustrates the co-occurrence probabilities estimated by different methods in the *kitchenware* scenario consisting of six classes: *bottle*, *cup*, *fork*, *knife*, *spoon*, *bowl*. We can see that CoP performs better than DiP on some classes, while performing worse on others. This means that although CoP benefits from the advantage of correlative prompts, it still suffers from the overfitting to co-occurrence. Suspicious co-occurrence hinders further performance improvement. To address this problem, DualPrompt aims to remove the negative impact caused by suspicious co-occurrence and achieve better co-occurrence estimation than both DiP and CoP.

## 6.6 TEXT-IMAGE RETRIEVAL RESULTS

To explore the mechanism behind DualPrompt, as shown in Figure 8, we visualize some images according to text-image similarity scores. Three rows represent the text-image retrieval results of DiP, CoP and DualPrompt. For each method, we illustrate the images ranked 1st, 5th, 10th, 15th, and 20th according to similarity scores. From the figure, we can see that the retrieved images from different methods exhibit significant differences. DiP tends to retrieve the images dominated by the target object. It sometimes makes a mistake due to the ambiguous object, *e.g.*, incorrectly identify a *vase* as a *cup* in the 15-th image. While CoP tends to retrieve images that depict scenes characterized by the target object and its co-occurring objects, such as a dining table full of plates or a fully equipped bathroom. This indicates that CoP identifies the target object by exploiting co-occurrence information. But it often suffers from the overfitting issue and makes incorrect predictions based on only the co-occurring objects, *e.g.*, there is no cup but only its co-occurring objects like *bowl* and

*bottle* in the 15-th image. The images retrieved by DualPrompt seem to be a combination of the first two methods. DualPrompt enhances the text-image matching ability by leveraging the advantages of the first two methods, which aims to identify the target object based on discriminative features and correlative patterns.

# 7 CONCLUSION

The paper studies zero-shot multi-label classification using CLIP without training. We find that compared to the single-label scenarios, CLIP often obtains unfavorable performance in the multi-label scenarios due to the fact that it does not explicitly exploit co-occurrence in both pretraining and inference. To address this problem, we propose the DualPrompt method to use the discriminative prompt and correlative prompt simultaneously. The former contains only the target label, which aims to capture discriminative features, while the latter contains the target label and its co-occurring labels, which introduces co-occurrence information. Moreover, we construct a causal inference framework to provide a theoretical explanation for our method. Extensive experimental results verify the effectiveness of the proposed method.

ACKNOWLEDGMENTS

MS was supported by JST ASPIRE Grant Number JPMJAP25B1.

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

Table 3: The comparative results between the proposed method and the state-of-the-art methods on Objects365 dataset. The best performance is highlighted in bold. Coo. Est. represents co-occurrence probabilities estimation.

| | Backbone | Resolution | Training Data Usage | mAP | F1 |
|---|---|---|---|---|---|
| CLIP | ResNet-101 | $224 \times 224$ | None | 30.0 | 26.8 |
| DualPrompt | ResNet-101 | $224 \times 224$ | 1% Data for Coo. Est. | 34.5 | 30.0 |
| TagCLIP | ViT-B/16 | $224 \times 224$ | None | 36.7 | 30.4 |
| DualPrompt + TagCLIP | ViT-B/16 | $224 \times 224$ | 1% Data for Coo. Est. | **37.9** | **31.9** |

## A  DERIVATION OF EQ. (2)

Considering that a correlative prompt $P^c$ consisting of the target label $L^t$ and its co-occurring labels $L^c$, let $\{L^t, L_0^c\}$ and $\{L_0^t, L^c\}$ represent two mutually exclusive events, where $L_0^c$ ($L_0^t$) represents we remove the co-occurring labels (target label) from the correlative prompt. By omitting the class index $k$ for notational simplicity, we have

$$
\begin{aligned}
&\mathcal{T}(\boldsymbol{x}) \\
&= p(y=1|\boldsymbol{x}, L^t, L^c) - p(y=1|\boldsymbol{x}, L^c, L_0^t) \\
&= p(y=1|\boldsymbol{x}, L^t, L^c) - \frac{p(y=1, \boldsymbol{x}, L^c, L_0^t)}{p(\boldsymbol{x}, L^c, L_0^t)} \\
&= p(y=1|\boldsymbol{x}, L^t, L^c) - \frac{p(y=1, \boldsymbol{x}, L^c, L^t) - p(y=1, \boldsymbol{x}, L_0^c, L^t)}{p(\boldsymbol{x}, L^c, L_0^t)} \\
&= p(y=1|\boldsymbol{x}, L^t, L^c) - \frac{p(y=1|\boldsymbol{x}, L^c, L^t)p(\boldsymbol{x}, L^c, L^t) - p(y=1|\boldsymbol{x}, L_0^c, L^t)p(\boldsymbol{x}, L_0^c, L^t)}{p(\boldsymbol{x}, L^c, L_0^t)} \\
&= (1 - \frac{p(\boldsymbol{x}, L^c, L^t)}{p(\boldsymbol{x}, L^c, L_0^t)})p(y=1|\boldsymbol{x}, L^t, L^c) + \frac{p(\boldsymbol{x}, L_0^c, L^t)}{p(\boldsymbol{x}, L^c, L_0^t)}p(y=1|\boldsymbol{x}, L_0^c, L^t) \\
&\propto p(y=1|\boldsymbol{x}, P^c) + \lambda p(y=1|\boldsymbol{x}, P^d),
\end{aligned}
\tag{3}
$$

where $\lambda = \frac{p(\boldsymbol{x}, L_0^c, L^t)}{p(\boldsymbol{x}, L^c, L_0^t) - p(\boldsymbol{x}, L^c, L^t)}$ can be regarded as a trade-off parameter. In our experiments, we find that our method achieves favorable performance by simply setting $\lambda$ to 1.

## B  COMPLETE RESULTS OF FIGURE 2(A)

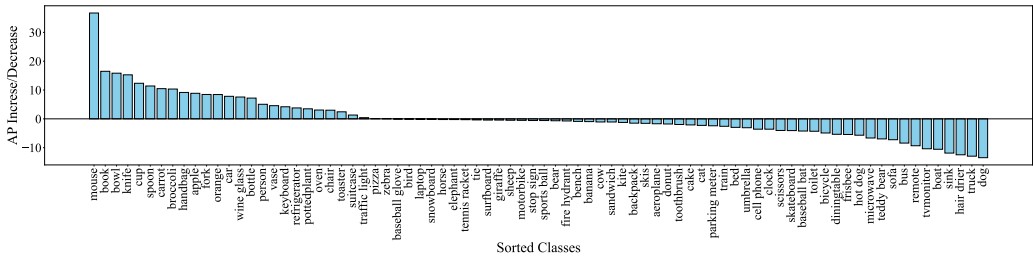

Figure 9: The performance changes of CoP against DiP in terms of AP over all classes. The performance of nearly half of the labels improved, while the performance of the other half decreased.

Figure 9 illustrates the performance changes of CoP against DiP in terms of AP over all classes. From the figure, we can see that nearly half of the labels experience performance improvements, while the other half suffer from performance declines. These results verify that co-occurrence has both good and bad effects.

## C    EXPERIMENTAL RESULTS ON OBJECTS365

To verify that the proposed method has a broad range of applicability, we have added experimental results on a more challenging dataset, Objects365 [4](Shao et al., 2019), which contains 365 classes. This dataset has an average of 6.17 labels per image, which is more than twice that of the COCO dataset (2.93). Table 3 reports the experimental results of our method and the comparison baselines on Objects365. For our method DualPrompt, we use 1% of the training data to estimate label co-occurrence. As shown in the table, DualPrompt significantly outperforms the baseline CLIP, achieving a 4.5% improvement in mAP. After incorporating TagCLIP, the proposed method still yields a 1.2% gain in mAP. These experimental results demonstrate the effectiveness of DualPrompt in challenging scenarios.

---

[4]https://www.objects365.org/overview.html

