# OpenReview forum: "Unlocking the Power of Co-Occurrence in CLIP: A DualPrompt-Driven Method for Training-Free Zero-Shot Multi-Label Classification"
_ICLR.cc/2026/Conference — ICLR 2026 Poster_

### Official Review · Reviewer_3XJU · 2025-10-20

**Soundness:** 3
**Presentation:** 3
**Contribution:** 2
**Rating:** 4
**Confidence:** 4

**Summary:**

The author tries to achieve zeroshot multi-label classification based on CLIP, and explore this task with labeling the object co-occurrence in CLIP. The paper improve the prompt to achieve the goal, and also calibrate CLIP predictions to achieve SOTA results. The experimental results are verified on mutiple benchmarks.

**Strengths:**

- This paper proposes a better prompt (discriminative prompt + correlative prompt) on the CLIP model (or similar models) to achieve better performance.

- The method only needs a few training data to derive the co-occurring labels.

**Weaknesses:**

- The statement in the abstract is a bit vague. it would be better to add a sentence to summarize what method you used to 'keeping the positive effect while removing the negative effect caused by suspicious co-occurrence'.

- The results seems promising on two datasets (MSCOCO and VG-256), but it would be great if the NUS-WIDE results can also be provided since it widely used with much larger dataset / number of labels and provide more convincing results.

**Questions:**

- In the introduction, the author mentioned that "There are two major challenges in achieving effective zeroshot multi-label classification", one is "CLIP tends to capture the global features of the dominated objects and neglecting the local features on otheres", another is "CLIP does not explicitly leverage label correlations". So I want to post two questions on these statements. 1). the first statement seems not like a challenge in achieving effective zeroshot multi-label classification for CLIP, since the author mentioned ViT can solve this issue while CLIP already have the ViT version, this paper's best model is also based on ViT. Why using ViT seems like a trouble in this paragraph, can you explain a bit more? 2). In the second challenge, the author shows an example of the co-occurrence probability, and demonstrate that CLIP cannot precisely model co-occurrence relationships, but it suddently comes to the conclusion that " resulting in unfavorable recognition performance". So why it results in unfavorable recognition performance? I think more explanation should be added here.

- In the experiment section, why the training data usage is 1% in MSCOCO and 2% for VG-256? How will this number influence the effectiveness of your method?

- Besides, I notice that this paper is also submitted in ICML 2025 and NeurIPS 2025 but without replying to any reviewer comments [[link](https://openreview.net/forum?id=2WLEEo4tqT)] [[link](https://openreview.net/forum?id=oCDPvJ2uyc)], could you simply summarize these questions and provide the answer?

---

> ### Author Response · Authors · 2025-11-20
>
> Thanks for your constructive comments. We are glad to answer all your questions.
>
> **Q1**: The statement in the abstract is a bit vague. it would be better to add a sentence to summarize what method you used to 'keeping the positive effect while removing the negative effect caused by suspicious co-occurrence'.
>
> **R1**: Thank you for the suggestion.  We have added the following sentence to the abstract:  “This can be achieved by using dual prompts consisting of the discriminative and correlative prompts, which introduce label co-occurrence while emphasizing the discriminative pattern of the target object.”.
>
> **Q2**: Why using ViT seems like a trouble in this paragraph, can you explain a bit more?  Why failure to exploit label correlations results in unfavorable recognition performance? I think more explanation should be added here.
>
> **R2**: In most cases, we prefer methods that can be applied to different models. Generally, a method with strong applicability implies broader usability across various scenarios. In some resource-constrained situations, for example, one may only be able to use convolution-based networks rather than ViT. In our first motivation, we aimed to emphasize that **one of the advantages of the proposed method** when compared with TagCLIP, is **its broad applicability across different models**.
>
> Compared with single-label classification, the output space of multi-label classification **grows exponentially** (that is, $2^q$, where $q$ is the number of classes). Therefore, effectively modeling label co-occurrence relationships, namely whether different labels tend to appear together, is an efficient way to reduce complexity. If a method fails to model label co-occurrence effectively, it indicates that important prior information has not been captured, which often results in missing labels and consequently degraded model performance. We have include the above discussion in the rebuttal version.
>
> **Q3**: In the experiment section, why the training data usage is 1% in MSCOCO and 2% for VG-256? How will this number influence the effectiveness of your method?
>
> **R3**: The VG-256 dataset contains 256 classes, which is larger than COCO with 80 classes. From a statistical perspective, VG-256 therefore **requires more instances to obtain a more accurate estimation of label co-occurrence probabilities**. In general, as the number of instances increases, the accuracy of estimating label co-occurrence probabilities improves. Once the instance size reaches a certain level, the estimation of co-occurrence probabilities becomes stable. The experimental results in Figure 6 confirm this observation. As shown in Figures 6(c) and 6(d), the performance of DualPrompt increases significantly when the amount of training data grows from 1% to 2%, while the subsequent improvements become relatively smaller. Table 5 reports the performance of DualPrompt in terms of mAP under different proportions of training data on VG-256.
>
> Table 5. Performance changes as the amount of training data used for estimating label co-occurrence increases
>
> | **Training Data Usage** |  1%  |  2%  |  3%  |
> | :---------------------: | :--: | :--: | :--: |
> |       **mAP (%)**       | 34.7 | 35.2 | 35.4 |

---

> > ### Author Response · Authors · 2025-11-20
> >
> > **Q4**: Besides, I notice that this paper is also submitted in ICML 2025 and NeurIPS 2025 but without replying to any reviewer comments [[link](https://openreview.net/forum?id=2WLEEo4tqT)] [[link](https://openreview.net/forum?id=oCDPvJ2uyc)], could you simply summarize these questions and provide the answer?
> >
> > **R4**: Since there are too many questions, starting from **Q5**，we selected several that the reviewers were particularly concerned about and that have not yet been addressed in this rebuttal for detailed responses. If there are any additional questions that are not covered in our rebuttal but are of particular interest to you, please let us know.
> >
> > **Q5**: It was not clear, whether the test data overlaps with CLIP’s pretraining distribution as this questions on how much it is depending on prior knowledge and co-occurrence.
> >
> > **R5**: From an experimental perspective, the performance of vanilla CLIP is relatively poor, showing a large gap compared with its improved variants such as TagCLIP and the proposed method. This suggests that there is little overlap between CLIP’s pre-training data and the test data used in our experiments. From the perspective of label co-occurrence, CLIP is trained on large-scale and diverse datasets, where the label co-occurrence patterns differ significantly from those in specific datasets such as COCO and VG-256 used in our study.
> >
> > **Q6**: Can the authors show how correlative prompts specifically target new objects not captured by DiP? For instance, are there cases where only CoP activates a label previously ignored by DiP?
> >
> > **R6**: Figure 8 visualizes several images based on text–image similarity scores obtained using DiP, CoP, and DualPrompt. From the figure, we can observe that DiP and CoP exhibit clearly **different recognition behaviors**. Specifically, DiP tends to favor images **dominated by the target object**, whereas CoP prefers images that depict scenes **characterized by the target object together with its co-occurring objects**. Since DiP focuses only on the target category, it may make mistakes when the object is ambiguous or difficult to recognize. In contrast, CoP can leverage the co-occurrence relationships among different categories to assist recognition and enhance overall performance. Please refer to **Section 6.6** for more detailed information.
> >
> > **Q7**: The paper does not clearly explain how the final multi-label predictions are derived from the predicted probabilities. While the approach for single-label classification is straightforward, multi-label classification typically requires a thresholding strategy to decide which labels to assign, and this is not discussed.
> >
> > **R7**: In our experiments, none of the methods included specific techniques for threshold selection. To ensure a fair comparison, for all methods, we sample a fixed proportion of the training data (1% for COCO 2014 and 2% for VG-256) and determine the threshold for each class according to its positive label proportion on the sampled data..
> >
> > **Q8**: The method alone does not achieve state-of-the-art performance. According to Table 1 and Table 2, the best results come from combining the proposed DualPrompt method with TagCLIP, rather than from DualPrompt alone. The absence of standalone results under comparable settings (Vit-B/16 backbone and original resolution) makes it difficult to evaluate the true effectiveness of the proposed method by itself.
> >
> > **R8**: This demonstrates the versatility of our method from another perspective. It functions as a plug-and-play module that can be integrated into any existing method to achieve significant performance improvements. When applying it to TagCLIP, it yields gains of **1.3%**, **1.1%**, and **1.2%** mAP on COCO, VG-256 and Objects365, respectively. These experimental results convincingly demonstrate the effectiveness of the proposed method.

---

> > > ### Author Response · Authors · 2025-11-20
> > >
> > > **Q9**: The paper's motivation isn't novel. Several existing methods, such as [1]-[4], already explore adding label correlation to CLIP prompts and addressing spurious label relationships.
> > >
> > > **R9**: I will summarize these methods and discuss their differences from our approach individually.
> > >
> > > [1] proposes Knowledge-Aware Prompting (KAP), which queries the inherent knowledge in large language models (LLMs) through in-context learning to generate informative descriptions. Although our method also operates on prompts, our idea aims to **introduce label co-occurrence relationships into the model** through prompting, rather than **enhancing recognition by generating more detailed object descriptions**.
> > >
> > > [2] employs causal inference theory to model label relationships. The main differences between their method and ours are as follows: 1) **The modeling of label relationships is fundamentally different**. [2] introduces a *confounder* to model label relationships, whereas we use a *mediator*. This leads to entirely different causal effects, which are also reflected in the causal graphs. 2) Due to the difference in modeling, [2] adopts the *Causal Intervention* technique, while we employ the *Total Direct Effect* approach. These are **two fundamentally different methodologies**.
> > >
> > > [3] also employs causal inference theory to address the multi-label classification problem. Compared with our method, in addition to the difference in how label relationships (referred to as *context knowledge* in their paper) are modeled, the techniques used are also different. The two methods are designed for entirely different scenarios: [3] focuses on the conventional multi-label classification setting that requires training data for model learning, while our method targets the zero-shot scenario, where no training data are used (a small amount of data is used only for estimating label co-occurrence, not for training). As a result, the implementations are fundamentally different. Specifically, [3] introduces causal effects through loss design during training, whereas our method, which does not modify model parameters, introduces label co-occurrence in a more elegant way through prompting.
> > >
> > > [4], similar to [2], also employs the causal intervention technique to learn causal label relationships for multi-label image classification. It is entirely different from our method in terms of label relationship modeling, technical approach, implementation, and application scenario.
> > >
> > > [1] Tan H, Tan Z, Li J, et al. SSPA: Split-and-Synthesize Prompting with Gated Alignments for Multi-Label Image Recognition[J]. arXiv preprint arXiv:2407.20920, 2024.
> > >
> > > [2] Chen Z M, Jin X, Chan S. In pursuit of causal label correlations for multi-label image recognition[J]. Advances in Neural Information Processing Systems, 2024, 37: 51634-51654.
> > >
> > > [3] Liu R, Liu H, Li G, et al. Contextual debiasing for visual recognition with causal mechanisms[C]//Proceedings of the IEEE/CVF Conference on Computer Vision and Pattern Recognition. 2022: 12755-12765.
> > >
> > > [4] Tian Y, Bai K, Yu X, et al. Causal multi-label learning for image classification[J]. Neural Networks, 2023, 167: 626-637.
> > >
> > > **Q10**: Why not use the complete training data to obtain co-occurrence, the choice of 1% and 2% of the data is not explained. It is not guaranteed that the low data will capture the true data distribution and hence the true co-occurrence.
> > >
> > > **R10**: Using the full set of training data to estimate label co-occurrence would violate the zero-shot setting, which typically assumes no training data. In our experiments, only a very small portion of training data is sufficient to estimate label co-occurrence. This can be verified by the results in Figure 6, where approximately 1% of the training samples on COCO and about 2% on VG-256 are sufficient to obtain satisfactory performance. For a fair comparison, we also compare our method with prompt tuning approaches (see Figure 6). The experimental results show that such a small amount of data is insufficient to fine-tune an effective model, while our method significantly outperforms these approaches, further demonstrating its practicality and effectiveness.

---

> > > > ### Comment · Reviewer_3XJU · 2025-11-22
> > > >
> > > > Thanks for the response! The author has addressed my major concerns and I would like to raise the score.

---

> > > > > ### Author Response · Authors · 2025-11-23
> > > > >
> > > > > We thank the reviewer for the positive response. We are glad that our clarifications addressed the reviewer’s concerns. We will further refine the discussions in the rebuttal and include them in the revised version. We sincerely appreciate the reviewer’s contribution to improving our paper.

---

### Official Review · Reviewer_PYxD · 2025-10-27

**Soundness:** 3
**Presentation:** 2
**Contribution:** 3
**Rating:** 4
**Confidence:** 4

**Summary:**

This paper proposes DualPrompt, a simple yet effective method for improving zero-shot multi-label classification with CLIP. By incorporating co-occurrence information into the prompt design via both Discriminative Prompt and Correlative Prompt, the authors aim to leverage label dependencies during inference without training. The paper provides theoretical justification through a causal inference framework and achieves competitive performance across standard benchmarks.

**Strengths:**

* The method is simple and elegant, requiring no fine-tuning and remaining compatible with standard CLIP backbones.

* The use of ChatGPT for co-occurrence estimation is a clever and pragmatic idea, enabling training-free augmentation of prompts.

* The paper is generally well-written, and the causal graph perspective helps to clarify the strengths and pitfalls of the method.

**Weaknesses:**

* Remaining Co-occurrence Issues Not Fully Analyzed:
Although the proposed method aims to mitigate overfitting caused by co-occurrence, it is likely that some object hallucination or over-reliance on correlations remains, especially when the co-occurring object is much more prominent than the target object. However, there is no in-depth analysis or diagnostic results to explore these residual errors in detail.

* Narrow Dataset Evaluation:
The experiments are conducted only on MS-COCO and VG-256. While VG-256 covers a wide label space, this is still limited compared to related works. Additional benchmarks such as VOC2007 or NUS-WIDE, which are widely used in multi-label classification, would strengthen the empirical evaluation and demonstrate the robustness and generalizability of the approach.

* Minor Issues:
There are a few editorial errors, e.g., the phrase “Eq. equation” appears, which should be corrected. Such issues, while minor, slightly detract from the overall polish of the paper.

**Questions:**

* Could you provide more detailed error analysis or qualitative cases showing the failure modes of DualPrompt, especially in cases where co-occurrence may still lead to false positives?

* Have you tried evaluating the method on other commonly used multi-label datasets (e.g., VOC2007, NUS-WIDE)? If not, is there any reason for excluding them?

* How sensitive is the model to the choice of co-occurring labels generated by ChatGPT? Are there failure cases when ChatGPT introduces incorrect or spurious co-occurrence?

**Details Of Ethics Concerns:**

None.

---

> ### Author Response · Authors · 2025-11-20
>
> Thanks for your constructive comments. We are glad that you considered our work “simple and elegant method, clever and pragmatic idea, generally well-written”. We are glad to answer all your questions.
>
> **Q1**:  Could you provide more detailed error analysis or qualitative cases showing the failure modes of DualPrompt, especially in cases where co-occurrence may still lead to false positives?
>
> **R1**: Thank you for the suggestion. In the rebuttal version, we have added a subsection titled *Failure Case Analysis* to analyze the failure cases of DualPrompt in addressing the overfitting problem to label co-occurrence. In these failure cases, we can observe that each image contains one or more salient objects that strongly co-occur with the target category, which leads DualPrompt to make incorrect predictions. Please refer to **Section 6.7** in the rebuttal version for more detailed information. However, we must emphasize that such cases are relatively rare for DualPrompt. In the majority of situations, DualPrompt effectively alleviates the overfitting problem caused by label co-occurrence.
>
> **Q2**: There are a few editorial errors, e.g., the phrase “Eq. equation” appears, which should be corrected. Such issues, while minor, slightly detract from the overall polish of the paper.
>
> **R2**: Thank you for the suggestions. We have carefully proofread the manuscript, corrected the language errors, and highlighted all the modifications.
>
> **Q3**: How sensitive is the model to the choice of co-occurring labels generated by ChatGPT? Are there failure cases when ChatGPT introduces incorrect or spurious co-occurrence?
>
> **R3**: The label co-occurrence provided by ChatGPT is often not perfectly consistent with the true label co-occurrence in the dataset, meaning that the generated co-occurrence information contains noise. We have also verified this phenomenon experimentally. In Table 1, the co-occurrence estimated using 1% of the data can be regarded as the ground-truth co-occurrence within the dataset. As shown, when using the co-occurrence relationships generated by ChatGPT, our method **DualPrompt (without using training data, denoted as “None”)** experiences a performance drop of 1.6% when using ResNet-101, respectively, compared to using co-occurrence estimated from 1% of the training data.
>
> However, we must emphasize that such a small amount of data (for example, **1% for COCO** and **2% for VG-256**) is insufficient for effectively fine-tuning CLIP. As shown in Figure 6, fine-tuning CLIP on such limited data results in significantly lower performance than our proposed method. This demonstrates that using a small portion of data to estimate **label co-occurrence** does not compromise the practical value of our approach.

---

> > ### Comment · Reviewer_PYxD · 2025-11-20
> >
> > Thank you for the detailed responses. I appreciate the clarifications and the additional analyses provided in the rebuttal version. I have a few follow-up questions and concerns regarding R1 and R3.
> >
> > Q1–R1 (Failure Case Analysis):
> >
> > In Figure 9 of Section 6.7, the failure cases appear to be instances where objects strongly co-occur with the target category, leading DualPrompt to make incorrect predictions. However, my understanding is that the primary motivation of the proposed method is precisely to mitigate overfitting caused by such co-occurrence biases.
> >
> > Given this, I am not fully convinced by the statement that “DualPrompt effectively alleviates the overfitting problem caused by label co-occurrence.” The shown failure cases seem to directly contradict this central claim, as they illustrate a core setting where one would expect DualPrompt to offer robustness.
> >
> > Q3–R3 (ChatGPT-generated Co-occurrence & Failure Cases):
> >
> > I acknowledge the results in Table 1 showing that ChatGPT-generated co-occurrence leads to a modest performance drop, and I had noted this previously. However, the response does not fully address my earlier question regarding **failure cases specifically arising from noisy or spurious co-occurrences generated by LLMs**.
> >
> > For many real-world applications, users may not have dataset access and may rely on automatically generated or heuristic co-occurrence knowledge (e.g., from LLMs). In such settings, it is important to understand:
> >
> > * **What kinds of incorrect or hallucinated co-occurrence relations introduced by ChatGPT can cause DualPrompt to fail**,
> > * Whether certain categories are more sensitive to spurious co-occurrence relations,
> > * And what practical guidance one might follow to design prompts or validate generated co-occurrences when dataset statistics cannot be computed.
> >
> > Even a minimal set of qualitative examples or taxonomy of failure types would significantly increase the practical utility of the work.
> >
> > Q - New question regarding the use of small portions of the dataset (e.g., 1% of COCO, 2% of VG-256):
> >
> > Were these subsets sampled completely at random?

---

> > > ### Author Response · Authors · 2025-11-21
> > >
> > > First, we would like to thank the reviewer PYxD for the prompt response. These questions are very insightful and closely related to our work. We believe that engaging in a in-depth discussion with the reviewer will help further improve our paper. We sincerely appreciate the reviewer’s time and effort.
> > >
> > > **Q1**:  The shown failure cases seem to directly contradict this central claim, as they illustrate a core setting where one would expect DualPrompt to offer robustness.
> > >
> > > **R1**: As we mentioned in our previous rebuttal (R1), “such cases are relatively rare for DualPrompt”, which indicates that there are far more cases in which DualPrompt succeeds. Although we present several failure examples of DualPrompt, these examples alone do not imply that **DualPrompt cannot effectively alleviate the overfitting problem**. Similarly, we could also present **many more successful examples**, but these would likewise not serve as definitive evidence that **DualPrompt can effectively alleviate the overfitting problem**.
> > >
> > > A direct way to evaluate the effectiveness of DualPrompt is to **compare it with CoP (Correlative Prompt)**. The table below presents the comparison results, from which we can clearly observe that **DualPrompt significantly outperforms CoP**. In addition, as shown in Figure 5(b) of the paper, we further provide the per-class performance improvements achieved by DualPrompt over CoP. The figure demonstrates that **DualPrompt improves performance on nearly all categories**. These experimental results demonstrate that **DualPrompt can effectively alleviate the overfitting problem**.
> > >
> > > |            | mAP  |
> > > | ---------- | ---- |
> > > | CoP        | 62.7 |
> > > | DualPrompt | 67.1 |
> > >
> > > **Q3-1**: What kinds of incorrect or hallucinated co-occurrence relations introduced by ChatGPT can cause DualPrompt to fail?
> > >
> > > **R3-1**: When we introduce co-occurrence relationships that are **inconsistent with the statistical co-occurrence of the dataset**, the performance of DualPrompt is negatively affected. This mainly happens in two situations:
> > >
> > > 1. **Introducing classes with low co-occurrence probability with the target class**, which can be regarded as *spurious co-occurring classes*. This leads to **object hallucination**, where the predicted probability of the target class becomes dominated by these spurious co-occurring classes.
> > > 2. **Failing to include classes that have high co-occurrence probability with the target class**, which can be viewed as *missing co-occurring classes*. In this case, DualPrompt is unable to fully leverage the true co-occurrence relationships that actually exist in the data.
> > >
> > > To examine the two cases discussed above, **Figure 11 in the rebuttal version** presents the per-class performance comparison of DualPrompt when using co-occurrence relationships generated by ChatGPT versus those estimated from data.
> > >
> > > For the first case, we observe some clear examples. For the **person** class, the co-occurring classes generated by ChatGPT, such as *bicycle* and *dog*, do not appear in the data-estimated co-occurring classes (which include *frisbee*, *tennis racket*, *baseball glove*, *baseball bat*, etc.).
> > > Similarly, for the **cat** class, the co-occurring classes generated by ChatGPT, such as *sofa* and *bed*, do not appear in the data-estimated co-occurring classes (which is empty because we require a co-occurrence probability greater than 0.2 to be considered valid). These **spurious co-occurring classes** provided by ChatGPT cause the model to produce **object hallucinations**, resulting in inferior performance compared with using data-estimated co-occurrence.
> > >
> > > For the second case, for the **fork** classes, several co-occurring classes identified through data estimation, including *diningtable*, *spoon*, *cup*, and *cake*, do not appear in the co-occurring classes provided by ChatGPT. Similarly, for the **book** class, the co-occurring classes estimated from data, such as *tvmonitor*, *chair*, and *sofa*, are missing from the co-occurring classes given by ChatGPT. These missing co-occurring classes **prevent the model from leveraging the true co-occurrence relationships** present in the dataset, which leads to decreased performance.

---

> > > > ### Author Response · Authors · 2025-11-21
> > > >
> > > > **Q3-2**: Whether certain categories are more sensitive to spurious co-occurrence relations?
> > > >
> > > > **R3-2**:  Classes that depend heavily on label co-occurrence, meaning they co-occur strongly with many other classes, are particularly sensitive to spurious co-occurrence relations. When a class has many co-occurring classes, it is more likely, compared with classes that have few co-occurring classes, that ChatGPT may **introduce more spurious co-occurring classes or omit more true co-occurring classes**. Both cases lead to a decrease in performance.
> > > >
> > > > As shown in Figure 11, for a group of frequently co-occurring categories such as the *kitchenware* group, which includes *bottle*, *wine glass*, *cup*, *fork*, *knife*, *spoon*, and *bowl* (their co-occurrence relationships can be found in Figure 1(d)), the data-estimated co-occurrence consistently leads to better performance than the ChatGPT-generated co-occurrence. A similar case can be observed for another group of frequently co-occurring categories, including *microwave*, *oven*, *toaster*, *sink*, and *refrigerator* (their co-occurrence relationships can be found in Figure 1(b)). This indicates that when a class has more co-occurring classes, it becomes more sensitive to spurious co-occurrence relationships and **can benefit more from incorporating data-estimated co-occurrence relationships to correct such errors**.
> > > >
> > > > **Q3-3**: What practical guidance one might follow to design prompts or validate generated co-occurrences when dataset statistics cannot be computed?
> > > >
> > > > **R3-3**: When dataset statistics cannot be computed, it often corresponds to two situations.
> > > >  The first situation is **when images are available but labels are not**. In this case, we can annotate only a very small portion of the images to estimate label co-occurrence probabilities. For example, in COCO 2014, the training set contains about 80k images, and annotating only 1%, that is, approximately 800 images, is sufficient. Annotating even fewer images is also effective, although the performance will decrease slightly.
> > > >
> > > > The second situation is **when neither images nor labels are available**. In this case, there are three possible approaches.
> > > >  The first approach is to consult domain experts. For instance, in the multi-label image classification tasks, one can ask experienced researchers to list the top $k$ categories that most frequently co-occur with each class, and then aggregate the responses from multiple experts.
> > > > The second approach is the one used in our work, which queries a chatbot like ChatGPT.
> > > > The third approach is to use a vision-language model such as CLIP, where category names are fed into the text encoder and the similarity between them is used as a surrogate for co-occurrence relationships.
> > > >
> > > > **Q4**: Were these subsets sampled completely at random?
> > > >
> > > > **R4**: Yes, these subsets were sampled completely at random. In fact, random sampling is necessary to ensure that the co-occurrence relationships within the subset remain consistent with those of the full dataset.

---

> > > > > ### Comment · Reviewer_PYxD · 2025-11-21
> > > > >
> > > > > Thank you for the detailed responses. My concerns have been fully resolved, and I will raise my rating to a 6.
> > > > >
> > > > > I do have one additional question. If hierarchical relationships among classes are well understood, can ChatGPT leverage such information effectively to further improve the co-occurrence estimation or the overall method?

---

> > > > > > ### Author Response · Authors · 2025-11-22
> > > > > >
> > > > > > In general, hierarchical relationships are helpful for identifying co-occurring classes, since subclasses under the same superclass are highly likely to appear together. For example, in the COCO dataset, subclasses under the *vehicle* superclass, such as *car*, *motorbike*, *bus*, and *truck*, frequently appear in the same image. Similar cases can be observed in many other supercategories as well, such as *kitchenware*, *appliance*, and *electronics*. If we provide these hierarchical relationships to ChatGPT as context information in the prompt, there is a high chance that they can help ChatGPT produce more accurate estimates of co-occurrence relationships.
> > > > > >
> > > > > > We would like to once again express our sincere gratitude to the reviewer for the constructive and insightful feedback, which has greatly contributed to strengthening this work. We will further improve these contents and include the experimental results and discussions in the revised version.

---

### Official Review · Reviewer_C23F · 2025-10-28

**Soundness:** 2
**Presentation:** 3
**Contribution:** 3
**Rating:** 6
**Confidence:** 3

**Summary:**

The paper targets zero-shot multi-label image classification with CLIP, noting that vanilla CLIP underperforms because it (i) focuses on the most salient object and misses others, and (ii) does not explicitly leverage label co-occurrence during pretraining or inference. The authors introduce correlative prompts (CoP) that append a target label with its frequent co-occurring labels inside the prompt, explicitly injecting co-occurrence information at inference time. They show CoP can boost recognition of less-salient objects but also induces object hallucination when correlated context appears without the target.

**Strengths:**

1. The method provides a simple, training-free mechanism to inject label co-occurrence at inference, which meaningfully improves zero-shot multi-label performance.
2. The approach is backbone-agnostic and complements patch-centric methods (e.g., TagCLIP), showing additive gains when combined.
3.The study includes informative diagnostics (class-wise AP changes, true/false positives, retrieval visualizations) that clarify how prompts alter model behavior.

**Weaknesses:**

1.The theoretical derivation from a TDE subtraction to a simple additive fusion lacks rigorous justification; the introduced λ parameter is set to 1 without sensitivity analysis.
2.The use of softmax across labels for multi-label probabilities and the summation of two scores raises calibration concerns and probabilistic interpretation remains unclear.
3.The evaluation scope is limited to two datasets and broader generalization and cross-dataset portability of co-occurrence priors are not assessed.
4.The method’s inference-time overhead should be reported (text encoding caching, similarity computation cost) for scalability considerations.
5.The comparisons should control for image resolution, prompt ensembling, and preprocessing; decision thresholds should be explicitly defined and consistently applied.

**Questions:**

None

---

> ### Author Response · Authors · 2025-11-20
>
> Thanks for your appreciation of our paper. We are glad that you considered our work “simple, meaningful improvement, informative diagnostics”. We are glad to answer all your questions.
>
> **Q1**: The theoretical derivation from a TDE subtraction to a simple additive fusion lacks rigorous justification; the introduced λ parameter is set to 1 without sensitivity analysis.
>
> **R1**: We have corrected the mistake and provided a clearer derivation in Appendix A. And we have added the corresponding reference in the main text. To use a single parameter $\lambda$ to control the contributions of the two terms, we rewrite the TDE formulation in Eq. (2) as $T_k(x) = (1 - \lambda)\text{Term 1} + \lambda\text{Term 2}$. This is a common trick in practice. Table 4 reports the performance variation as $\lambda$ changes. From the table, we can observe that the model **achieves the best performance when $\lambda = 0.5$**, meaning that the two terms contribute equally. This indicates that both the label co-occurrence information and the discriminative features play equally important roles.
>
> Table 4: Performance changes with respect to the parameter $\lambda$.
>
> | $\lambda$ |  0.3  |  0.4  |  0.5  |  0.6  |  0.7  |
> | :-------: | :---: | :---: | :---: | :---: | :---: |
> |    mAP    | 66.69 | 67.00 | 67.14 | 67.05 | 66.87 |
>
> **Q2**: The use of softmax across labels for multi-label probabilities and the summation of two scores raises calibration concerns and probabilistic interpretation remains unclear.
>
> **R2**: In multi-label settings, the Sigmoid function is typically used as the activation function. However, as demonstrated in [1], when using CLIP for inference, **applying Sigmoid instead of Softmax leads to a significant drop in performance** (see Table 2 in [1]). This may be because CLIP itself is trained with Softmax as the activation function.
>
> For the summation operation, we performed calibration by dividing by 2 in practical implementation.
>
> In Eq. (2), $p(y_k = 1 \mid x, P_k^c)$ denotes the model output for the $k$-th class using the correlative prompt, while $p(y_k = 1 \mid x, P_k^d)$ denotes the model output for the $k$-th class using the discriminative prompt.
>
> [1] TagCLIP: A Local-to-Global Framework to Enhance Open-Vocabulary Multi-Label Classification of CLIP Without Training.
>
> **Q3**: The method’s inference-time overhead should be reported (text encoding caching, similarity computation cost) for scalability considerations.
>
> **R3**: Thank you for the suggestion. For the entire dataset, the text encoding is **performed only once and can therefore be ignored**. The inference time mainly consists of **image encoding and similarity computation**. For both CLIP and our method DualPrompt, the image encoding time is identical. The only difference is that our method **requires two similarity computations**. However, compared with image encoding, which involves forward propagation, similarity computation is merely a simple matrix multiplication operation with negligible computational cost. As a result, **the overall inference time of our method is almost the same as that of CLIP**. The table below reports the time, measured in seconds, required for image encoding, CLIP, and DualPrompt to perform inference on a batch of 64 images.
>
> | Method         | Time Cost (S) |
> | :------------- | :-----------: |
> | Image encoding |    2.24330    |
> | CLIP           |    2.26944    |
> | DualPrompt     |    2.26977    |
>
> **Q4**: The comparisons should control for image resolution, prompt ensembling, and preprocessing; decision thresholds should be explicitly defined and consistently applied.
>
> **R4**: To ensure a fair comparison, we keep all method-independent settings, inlcuding image resolution, prompt template, and preprocessing, the same across different methods. Since none of these methods focus on the thresholding issue, for all methods, we sample a fixed proportion of the training data (1% for COCO 2014 and 2% for VG-256) and determine the threshold for each class according to its positive label proportion on the sampled data.

---

> > ### Comment · Reviewer_C23F · 2025-11-28
> > **Thanks for the response**
> >
> > All my concerns have been addressed well, and I decide to raise my score to 8.

---

> > > ### Author Response · Authors · 2025-11-28
> > >
> > > Thank you very much for your response. We appreciate the time and effort you have devoted to reviewing our paper. We will include these constructive suggestions in the revised version.

---

> ### Comment · Area_Chair_jcEx · 2025-11-28
> **Please reply to the authors' rebuttal**
>
> Dear Reviewer,
>
> The authors have provided their rebuttal. Please reply to it before the rebuttal period ends. Thanks!
>
> Best regards,
>
> AC

---

### Official Review · Reviewer_QsUa · 2025-11-01

**Soundness:** 3
**Presentation:** 2
**Contribution:** 2
**Rating:** 6
**Confidence:** 5

**Summary:**

This paper addresses the limitation of CLIP in zero-shot multi-label classification, where the model fails to explicitly leverage label co-occurrence relationships, leading to suboptimal performance. This paper proposes a novel method called DualPrompt, which introduces co-occurrence information by using two types of prompts: a discriminative prompt containing only the target label and a correlative prompt that includes both the target and its co-occurring labels. While the correlative prompt enhances recognition by capturing label dependencies, it also causes object hallucination due to overfitting to co-occurrence patterns. To mitigate this, this paper develops a causal inference-based calibration mechanism that retains the benefits of co-occurrence while suppressing its adverse effects. Extensive experiments on MS-COCO and VG-256 datasets demonstrate that DualPrompt outperforms both training-based and training-free state-of-the-art methods, achieving superior performance even without any training data.

**Strengths:**

1.	Proposes a novel, training-free method (DualPrompt) that explicitly injects label-co-occurrence information into CLIP for zero-shot multi-label classification.
2.	Introduces a lightweight dual-prompt strategy—discriminative vs. correlative—that can be applied to any CLIP backbone without training.
3.	Provides a principled causal-inference formulation that disentangles the helpful mediated effect of co-occurrence from the harmful “object-hallucination” effect, yielding calibrated predictions.
4.	Demonstrates consistent gains over both training-free (CLIP, TagCLIP) and training-based (DualCoOp, TaI) competitors on MS-COCO and VG-256.
5.	Shows that only ~1 % of training data is sufficient to estimate reliable co-occurrence statistics, keeping data requirements minimal.
6.	Validates interpretability via visualization: retrieved images align better with true co-occurrence patterns while suppressing false positives, confirming that the method preserves semantic plausibility.

**Weaknesses:**

1.	The paper only conducts experiments on MS-COCO and VG-256 datasets, which is not sufficient to demonstrate the generalizability of the proposed method. It is recommended to include experiments on more commonly used datasets in multi-label learning to further validate the effectiveness and robustness of the approach.
2.	The paper lacks discussion of recent works that also exploit label co-occurrence, e.g., SST [1] and SCPNet [2]. Although these papers address different settings, a comparative summary of how their co-occurrence modeling differs from DualPrompt is necessary to clarify novelty and avoid overlap.
3.	The paper places RELATED WORKS after the experiments and just before CONCLUSION, which breaks the usual flow and forces readers to assess the method’s novelty without early context.
4.	The derivation of Eq. (3) contains errors. Please carefully re-check every step.
5.	Table 1 reports TaI-CLIP with ResNet-101 trained on COCO Captions, but the corresponding ViT-B/16 entry under the same training condition is missing.
6.	The manuscript contains numerous typographical and consistency errors that must be thoroughly corrected: in Figure 2 “false negatives” is mislabeled, “prompr” should be “prompt”, “Eq. equation” is redundant, the tables mix “x” and “×”, the caption of Figure 5 is wrongly copied from Figure 6, Section 5.5 lists “microwave, oven, toaster, sink, refrigerator” while Figure 7 shows “bottle, cup, fork, knife, spoon, bowl”, the reference “Counterfactual reasoning for multi-label image classification via patching-based training” is duplicated, and “cane” should be “can”.
7.	On line 201 the symbol F^t appears in the causal path L^t → F^t → Y without any prior definition; please clarify what this feature vector represents and how it differs from F^d and F^c to avoid confusing the reader.

[1] Chen, Tianshui, et al. “Structured semantic transfer for multi-label recognition with partial labels.” Proceedings of the AAAI conference on artificial intelligence. Vol. 36. No. 1. 2022.

[2] Ding, Zixuan, et al. “Exploring structured semantic prior for multi label recognition with incomplete labels.” Proceedings of the IEEE/CVF Conference on Computer Vision and Pattern Recognition. 2023.

**Questions:**

In Figure 6, why does DualCoOp++ (which presumably augments DualCoOp with additional parameters) yield lower mAP/F1 than DualCoOp?

---

> ### Author Response · Authors · 2025-11-20
>
> Thanks for your appreciation of our paper. We are glad that you considered our work “novel, principled formulation”. We are glad to answer all your questions.
>
> **Q1**: The paper lacks discussion of recent works that also exploit label co-occurrence, e.g., SST [1] and SCPNet [2]. Although these papers address different settings, a comparative summary of how their co-occurrence modeling differs from DualPrompt is necessary to clarify novelty and avoid overlap.
>
> **R1**: Thank you for the suggestion. We summarize these two papers and highlight how they differ from our method. To address the issue that dataset-level statistical correlations may not hold for every individual image, SST introduces the Intra-image Semantic Transfer (IST) module, which learns an image-specific co-occurrence matrix and leverages this information to recover unknown labels. To overcome the challenge that incomplete annotations make it difficult to estimate label co-occurrence, SCPNet computes text features by feeding category names into the CLIP text encoder, and uses their pairwise similarities as a surrogate for the label co-occurrence. Unlike these methods that focus solely on exploiting label correlations, our approach introduces correlative prompts to incorporate label co-occurrence into CLIP’s inference process, while simultaneously employing discriminative prompts to mitigate the overfitting issue to label correlations. We have included the above discussion in the rebuttal version.
>
> **Q2**: The paper places RELATED WORKS after the experiments and just before CONCLUSION, which breaks the usual flow and forces readers to assess the method’s novelty without early context.
>
> **R2**: Thank you for the suggestion. To enhance the readability of the paper, we have relocated the *Related Works* section to immediately follow the *Introduction* in the rebuttal version.
>
> **Q3**: The derivation of Eq. (3) contains errors. Please carefully re-check every step.
>
> **R3**: Thank you for the reminder. We have corrected the error and rewritten Eq. (3) accordingly in the rebuttal version.
>
> **Q4**: Table 1 reports TaI-CLIP with ResNet-101 trained on COCO Captions, but the corresponding ViT-B/16 entry under the same training condition is missing.
>
> **R4**: Following TagCLIP [1], we use the results of TaI reported in its original paper. The paper did not report results on ViT-B/16. It is also worth noting that a direct comparison with TaI (trained on COCO Captions) may not be essential for our method, as our approach requires only a very small amount of labeled data to estimate label co-occurrence. Alternatively, for ViT-B/16, we report the TaI results trained on 1% of the training data, which were used for estimating label co-occurrence in our method.
>
> [1] TagCLIP: A Local-to-Global Framework to Enhance Open-Vocabulary Multi-Label Classification of CLIP Without Training.
>
> **Q5**: The manuscript contains numerous typographical and consistency errors that must be thoroughly corrected: in Figure 2 “false negatives” is mislabeled, “prompr” should be “prompt”, “Eq. equation” is redundant, the tables mix “x” and “×”, the caption of Figure 5 is wrongly copied from Figure 6, Section 5.5 lists “microwave, oven, toaster, sink, refrigerator” while Figure 7 shows “bottle, cup, fork, knife, spoon, bowl”, the reference “Counterfactual reasoning for multi-label image classification via patching-based training” is duplicated, and “cane” should be “can”.
>
> **R5**: Thank you for the suggestions. We have carefully proofread the manuscript, corrected the language errors, and highlighted all the modifications in the rebuttal version.
>
> **Q6**: On line 201 the symbol F^t appears in the causal path L^t → F^t → Y without any prior definition; please clarify what this feature vector represents and how it differs from F^d and F^c to avoid confusing the reader.
>
> **R6**: Thank you for the reminder. The symbol $F^t$ is a writting error. The correct formulation should be $L^t → F^d → Y$. This pathway indicates that the label $L^t$ activates the corresponding discriminative features $F^d$, which in turn produce the prediction $Y$.
>
> **Q7**: In Figure 6, why does DualCoOp++ (which presumably augments DualCoOp with additional parameters) yield lower mAP/F1 than DualCoOp?
>
> **R7**: This may be because DualCoOp++ contains more learnable parameters than DualCoOp, and therefore requires more data for effective training. As more training data becomes available, we can observe that the performance gap between DualCoOp++ and DualCoOp gradually narrows

---

> ### Comment · Area_Chair_jcEx · 2025-11-28
> **Please reply to the authors' rebuttal**
>
> Dear Reviewer,
>
> The authors have provided their rebuttal. Please reply to it before the rebuttal period ends. Thanks!
>
> Best regards,
>
> AC

---

### Author Response · Authors · 2025-11-20

**Q**: Experiments on datasets beyond COCO 2014 and VG-256

**R**: To verify that the proposed method has a broad range of applicability, we have added experimental results on a more challenging dataset, Objects365, which contains 365 classes. This dataset has an average of **6.17 labels per image**, which is **more than twice** that of the COCO dataset (2.93). Table 3 reports the experimental results of our method and the comparison baselines on Objects365. For our method DualPrompt, we use 1% of the training data to estimate label co-occurrence. As shown in the table, DualPrompt significantly outperforms the baseline CLIP, achieving a **4.5% improvement in mAP**. After incorporating TagCLIP, the proposed method still yields a **1.2% gain in mAP**. These experimental results demonstrate the **effectiveness of DualPrompt in challenging scenarios**. We have included these result in the rebuttal version (see Appendix C) and will continue to improve the experiments on this dataset in the future version.

Table 3: The comparative results between the proposed method and the state-of-the-art methods on Objects365 dataset.

|                    |  Backbone  | mAP  |  F1  |
| ------------------ | :--------: | :--: | :--: |
| CLIP               | ResNet-101 | 30.0 | 26.8 |
| DualPrompt         | ResNet-101 | 34.5 | 30.0 |
| TagCLIP            |  ViT-B/16  | 36.7 | 30.4 |
| DualPrompt+TagCLIP |  ViT-B/16  | 37.9 | 31.9 |

Regarding the VOC 2007 and NUS-WIDE datasets, their **co-occurrence characteristics are relatively weak**. For example, in VOC 2007, **the average number of labels per image is only 1.44**, meaning that most images contain only a single label and almost no co-occurring objects. Although NUS-WIDE has an average of **2.43 labels per image**, this number is inflated by **hierarchical labeling**; for instance, *dog* and *animal* are counted as two labels even though they refer to the same object in the image. These factors make both datasets less suitable for evaluating co-occurrence relationships, which are **central to the motivation** of our paper. Compared with these datasets, Objects365 provides a **more comprehensive benchmark** for assessing zero-shot multi-label image classification, both in terms of the **size of label space** and the **average number of labels per image**.

---

### Author Response · Authors · 2025-12-02
**Rebuttal Summary**

Dear AC and SAC,

We would like to sincerely thank you for the time and effort devoted to handling our submission. To help reduce your workload, we provide the following rebuttal summary.

*We are glad that the reviewers appreciated and highlighted several contributions of our work:*

- **Novel, simple and elegant methodology** (Reviewer QsUa, C23F, PYxD): We propose a training-free multi-label image classification method that employs dual prompts consisting of a discriminative prompt and a correlative prompt, which together introduce label co-occurrence information while emphasizing the discriminative pattern of the target object. The method is plug-and-play and incurs almost no additional inference overhead compared to vanilla CLIP.

- **Solid theoretical justification through a principled causal inference formulation** (Reviewer QsUa, PYxD): We provide a causal inference framework that explains why the proposed method can alleviate the co-occurrence overfitting problem.

- **Informative experimental results** (Reviewer QsUa, C23F, PYxD, 3XJU): Comprehensive experimental results, including comparative experiments, ablation studies, and visualization analyses, validate both the effectiveness of the proposed method and its underlying working mechanism.

*During the rebuttal phase, we engaged in in-depth discussions with the reviewers and addressed their concerns. The details are as follows:*

- **Experiments on datasets beyond COCO and VG-256**: We additionally conducted experiments on the Objects365 dataset, which is a more challenging benchmark containing 365 classes and an average of 6.17 labels per image. The results show that our DualPrompt method outperforms the baseline CLIP and the SOTA method TagCLIP by 4.5% and 1.2% mAP, respectively, which demonstrates its effectiveness in challenging scenarios. Please refer to Appendix C for more details.

- **The issue on the derivation of Eq. (3)**: We rewrote the derivation of Eq. (3) and added the missing reference to Eq. (3) in the main text.

- **Sensitivity analysis of the hyper-parameter $\lambda$**: We conducted a sensitivity analysis of the parameter $\lambda$ in Eq. (3) and found that our method performs best when $\lambda = 0.5$, i.e., when the two terms contribute equally. This indicates that both the label co-occurrence information and the discriminative features play equally important roles.

- **Analysis of the failure cases of the proposed method**: We reported the overall experimental results and per-class performance comparison between DualPrompt and CoP (Correlative Prompt), showing that DualPrompt significantly outperforms CoP. These results demonstrate that DualPrompt can effectively address the co-occurrence overfitting problem.

*The reviewers acknowledged that their concerns had been addressed and provided positive feedback.*

**Reviewer PYxD**: Rating: 4 $\rightarrow$ 6 (see PYxD's comment *"I will raise my rating to a 6"*)

**Reviewer 3XJU**: Rating: 4 $\rightarrow$ 6 (see 3XJU's comment *"I would like to raise the score"*)

**Reviewer C23F**: Rating: 6 $\rightarrow$ 8 (see C23F's comment *"I decide to raise my score to 8"*)

Finally, we would like to once again express our sincere gratitude to you for the time and effort devoted to handling our submission. We would be extremely grateful if you could take the summary into consideration when making the final decision.

Best regards,
The Authors

---

### Meta-Review · Area_Chair_v1Yo · 2025-12-24

**Summary:**

This paper proposes a training-free approach for multi-label zero-shot learning based on CLIP. The authors contend that existing text prompts neglect co-occurrence relationships among labels. To address this, they introduce an intuitive method that incorporates the category label with the highest co-occurrence with the target label into the text prompt, thereby enhancing model performance. Experiments conducted across multiple datasets and baselines demonstrate significant improvements.

The paper gets an initial score of 6, 6, 4, and 4. Reviewers note that this paper presents a simple yet innovative approach. Primary concerns include:
1. Adding more experiments and analyses, including different datasets and computational costs.
2. Correcting some grammatical and formula errors.
3. Providing more detailed explanations for certain statements.

This paper does not exhibit significant fatal flaws. Additionally, the authors have proactively addressed certain concerns/questions in their response, and meanwhile, three reviewers have mentioned increasing the ratings in their follow-up comments. Given the above summary, I recommend acceptance.

**Reviewer Concerns:**

The authors have addressed certain concerns and questions raised by the reviewers. The main changes are as follows:
1. Additional comparative and ablation experiments. The authors have incorporated experiments using the Object365 dataset. Furthermore, they have included several ablation studies and analyses.
2. Correction of grammatical and formula errors. The authors have addressed and rectified the grammatical and formula errors identified by the reviewers.
3. Enhanced explanation of the motivation. The authors have provided a more detailed explanation of the limitations of previous methods and the rationale behind this work.

**Reviewer Scores:**

I think the authors have reasonably addressed the reviewers' concerns and questions in the response. Therefore, the reviewers may consider raising their scores.

---

### Decision · Program_Chairs · 2026-01-26

Accept (Poster)